# The effect of mothers and caregivers' fasting status on the dietary diversity of children 6-23 months: A longitudinal study in Debrebirhan, Ethiopia

**Addisalem Zebene Armdie** *, **Esete Habtemariam Fenta, Solomon Shiferaw**

Department of Nutrition and Dietetics, School of Public Health, Addis Ababa University, Addis Ababa, Ethiopia

* zebene.addisalem@gmail.com

**Data Availability Statement:** All relevant data are within the paper and its Supporting Information files.

## Abstract

### Background

There are various religions in Ethiopia, of which the Orthodox Tewahido Christian accounts for 44% of the population. According to the Ethiopian Orthodox Tewahido practice close to 200 days annually are dedicated to fasting. During this time, all followers who are above seven years old are expected to abstain from all types of food, including animal source foods and water for up to some hours daily. It is possible that such practice by mothers or caregivers could affect children's dietary practice. However, whether mothers/caregivers' fasting status influences dietary diversity of children during these periods remained uninvestigated.

### Methods

A community-based longitudinal study was conducted in Debrebirhan, North Shewa Zone, Ethiopia in seven randomly selected kebeles. We collected data in a sample of 218 mothers/caregivers, from January 29 to February 25, 2019 in the pre-fasting period and from March 18 to April 10, 2019, during fasting period on same participants. Data was entered on Epi-Data version 4.4.2.1 and analyzed using STATA 15 software. Children's dietary diversity was measured using the World Health Organization (WHO) standardized questionnaire for infant and young child feeding. The McNemar paired test was used for comparison of baseline and end line measurements. Statistical significance was set at p<0.05.

### Result

A total of 218 and 216 mothers/caregivers with children 6–23 months participated in the study before and during fasting season with a response rate of 100.0% and 99.0% respectively. The median age of children was 14 months. The proportion of children who met the minimum dietary diversity before the fasting season was significantly higher (23.4%) compared to during the fasting period (5.5%). (P<0.001). The proportion of children who consumed dairy product was significantly higher (55.5%) before the fasting period compared to

**Funding:** Funding for this research was provided by Addis Ababa University School of Public Health, The funders had no role in study design, data collection and analysis, publication decision, or manuscript preparation.

**Competing interests:** The authors have declared that no competing interests exist.

consumption during the fasting period (42.6%) (p<0.001). Similarly, consumption of flesh food was significantly higher before the fasting period (17.9%) compared to consumption during the fasting period (0.46%) (P<0.001).

## Conclusion

The study revealed that mothers/caregivers' fasting status negatively affect the dietary diversity of children aged 6–23 months in the household by decreasing their consumption of animal source food. Intervention strategies in promoting children's dietary diversity should be designed in a way that considers Ethiopian Orthodox Tewahido Christian mothers/caregivers' fasting practice.

## Introduction

The years from birth to two years of age is a critical period for infants and young children to have proper feeding practices in order to prevent child malnutrition [1]. Dietary diversity (DD) is an indicator used to assess the appropriateness of these infant and young child feeding (IYCF) practices. It refers to increasing consumption of the number of variety of foods across and within the food groups capable of ensuring adequate intake of essential nutrients. According to World Health Organization (WHO) recommendation, "minimum dietary diversity" (MDD) is defined as the proportion of children 6–23 months who consumed a minimum of four foods out of the seven food groups in the previous day. These food groups are grains, roots and tubers; legumes and nuts; dairy products; flesh foods; eggs; vitamin A-rich fruits and vegetables; and other fruits and vegetables [2].

However, attaining adequate dietary diversity for children 6 to 23 months has been a continual struggle for many countries around the world, particularly in South Asia and Sub-Saharan Africa [3] where children, 6–23 months with adequate minimum dietary diversity scores ranged from 15–71% [4, 5]. Ethiopia is also one of the countries with a high burden of inadequate dietary diversity, where only 14% of children aged 6–23 months meet the minimum dietary diversity requirement [6]. Individual, household, and societal factors have all been identified as contributing to inadequate dietary diversity thus far [7]. One sociocultural component that might influence individual dietary choices, household food consumption patterns, and family feeding practices is religion [8].

Ethiopia is home to a diverse range of religions, with Orthodox Tewahido Christians accounting for 44% of the population [9]. Fasting is an essential dietary regulation that the Ethiopian Orthodox Tewahido church follows to guide followers on what to eat and what not to eat [10]. It is defined as a partial or total abstention from all foods, or a select abstention from prohibited foods of animal origin for a limited time until the period of fasting is over. There are seven principal fasting periods annually. These are, 55 days lent fast period preceding Easter, the fast of the apostles ranging 14–44 days, the Fast of the Prophets of 43 days, the Fast of the Assumption 15 days in august, 3 days of the Fast of Nineveh, Tsome Gehad of Christmas and Epiphany (one day of fasting before Christmas and one day of fasting before Advent) and with the exception of two months after Ethiopian Easter, every Wednesday and Friday are fasting days almost all year-round. The great Easter fast (Lent) also known as Abiy Tsom is the most important and longest continuous fasting period of all fasts of the year [11].

Fasting for more than 200 days a year is required of all Orthodox Tewahido religion followers above the age of seven. Children under the age of seven, military personnel, the sick,

pregnant and nursing mothers are exempt from strict fasting and are allowed to eat both animal and non-animal source foods, as well as water, during the religious fasting periods [12].

Although mothers are well aware that their children are not required to fast, it is likely that adults in the household's fasting behavior affects children's access to a variety of foods, including foods of animal origin. This is owing to the fact that during fasting periods, women may not prepare non-fasting foods separately for their children for fear of touching the forbidden foods or contaminating tools used for family food cooking, as well as the lack of non-fasting foods in the market [13, 14]. This can possibly contribute to inadequate dietary diversity among children 6 to 23 months of age.

Despite concerns over the potential negative effect of fasting periods, there are limited published works investigating the effect of mothers/caregivers' fasting during the widely observed Ethiopian Orthodox Tewahido Christians' fasting periods on children's dietary diversity. Various studies have focused on other factors related to the child, mothers, or household characteristics; however, only a small number of papers have considered the impact of religious beliefs or fasting on child dietary diversity. The few available studies did not assess children's dietary diversity before and during fasting periods, making inference difficult [13, 14]. The objective of the present community-based longitudinal study was therefore to estimate the effect of mothers/caregivers' fasting practices during the Ethiopian Orthodox Lent fasting season on dietary diversity of children aged 6–23 months in Debrebirhan, Ethiopia.

## Materials and methods

### Study setting, design and population

We conducted a longitudinal study from January 29 to February 25, 2019 during the official pre-fasting period and from March 18 to April 10, 2019 during the official fasting period. The study was conducted in Debrebirhan woreda, which is located in North Shewa Zone, Amhara Regional State, 130 km away from Addis Ababa.

According to 2018 estimates, the wereda or district has a total population of 108,825. Within this population, the number of children aged 6–23 months are estimated to be 5,952, of which 5,084 reside in urban kebeles and 868 in rural kebeles (smallest administration units) [9].

Children 6–23 months whose mothers/care givers were Orthodox Tewahido Christian and who were living in randomly selected households were considered as the study population. Same households were visited and same mothers with children 6 to 23 months of age were interviewed in both rounds; before and during the fasting periods. Children whose mothers or caregivers were seriously ill or unable to hear and speak, who were absent on two repeated visits, and who had been ill in the past 1 week were excluded from the study.

### Sampling method

The required sample size was determined with the aid of statistical calculation program Epi Info version 7, using a double population proportion formula assuming the proportion of children with adequate dietary diversity to be 27.2% and 13.6% before and during fasting period, respectively [13], with 80% power, 95% confidence level, 10% non-response rate and 1.5 design effect. A correction formula was used and we have estimated the final sample size as 218. We have selected five and two kebeles randomly from urban and rural sites respectively. Children aged 6–23 months whose mothers were orthodox Christian fulfilling all the inclusion criteria were included by lottery method.

## Study procedure and measurements

We used an interviewer administered questionnaire to collect the required information from the mothers/caregivers. In the pre-fasting period, respondents were asked for their intention to fast in the coming lent fast period. Fasting generally entails eating only one meal per day, either in the evening or after 2.45 p.m., and abstaining from meat, fats, eggs, and dairy products [11]. However, some adhere strictly to this fasting rule, while others can fast (abstain from eating/drinking anything) for a few hours (9am or 12pm, for example), or some only fast from eating animal foods but may not stay up for several hours without taking any foods/drinks. Mothers/caregivers who abstained from all animal source foods or had stayed up to some hours without consuming any water or food in the previous day of the data collection period were considered as fasting during the lent fasting period. Children's dietary diversity was measured using the WHO Infant and Young Child Feeding Practices (IYCF) indicators. Children who consumed four or more food groups from the seven food groups (grain, roots and tubers, legumes and nuts, dairy products, meat, egg, vitamin A rich fruits and vegetables and other fruits and vegetables) in the past 24 hour before the interview were categorized as fulfilling the minimum dietary diversity (MDD) requirement [15]. The data collection tool was first prepared in English and then translated into Amharic prior to administration. Five health extension workers (HEWs) and two supervisors were recruited and trained about the use of the data collection tool. We interviewed the same group of participants before and during fasting season through house-to-house survey. The investigators along with two supervisors supervised the overall data collection process.

## Data analysis

The collected data was entered into Epi Data 7 after checking its completeness. Then it was exported to STATA 15 for cleaning and further analysis. Results from descriptive analysis were presented using tables in terms of numbers, percentage, mean, median, range and standard deviation. The McNemar paired test was used to compare the MDD before and during the fasting time and statistical significance was set at $p < 0.05$.

## Ethical considerations

Ethical clearance was obtained from an institutional review board within the University of Addis Ababa. A letter of support was obtained from Addis Ababa University to the Debrebirihan City Administration and Health Office. The study participants were informed about the purpose and procedures of the study. They were also told about their right to participate or not participate in the study. A written informed consent was obtained from the mothers/caregivers in order to be part of the study. We have used code to ensure privacy and confidentiality of study participants. Counseling was given to the mothers/caregivers whose children showed a significant difference in their dietary diversity during the two periods.

# Result

## Socio demographic, maternal and child characteristics

The study included 218 children aged 6–23 months, 52.7 percent of whom were males and the rest were females. The children were 14 months old on average. The majority (87.6 percent) of the mothers who participated in the study were married. The mothers or caregivers were 28.6 (SD 0.3) years old on average. One hundred twenty-five (57.4 percent) of the mothers had completed secondary school, and 60.5 percent were housewives. In terms of breast-feeding status, 88.5 percent of mothers were breastfeeding at the time of data collection (See Table 1).

**Table 1. Socio-demographic, maternal and child characteristics (n = 218) at baseline (Before fasting period) in Debrebirhan, North Shewa Zone Ethiopia (February-June, 2019).**

| Characteristics | Frequency (%) |
|---|---|
| **Residence** | |
| Urban | 172 (78.9%) |
| Rural | 46 (21.1%) |
| **Age of the mothers/caregivers** | |
| 15–24 | 33 (15.1%) |
| 25–34 | 155 (71.1%) |
| > = 35 | 30 (13.8%) |
| **Mother/caregivers education status** | |
| No formal education | 34 (15.6%) |
| Primary school | 59 (27.1%) |
| Secondary school | 78 (35.8%) |
| Higher education | 47 (21.6%) |
| **Fathers education status** | |
| No formal education | 44 (20.2%) |
| Primary school | 53 (24.3%) |
| Secondary school | 51 (23.4%) |
| Higher education | 70 (32.1%) |
| **Mothers/caregivers occupation** | |
| Housewife | 132 (60.5%) |
| Government employee | 35 (16.1%) |
| Merchant | 13 (5.9%) |
| Private/NGO | 33 (15.1%) |
| Other[a] | 5 (2.3%) |
| **Father occupation** | |
| Farmer | 38 (17.4%) |
| Government employee | 72 (33%) |
| Merchant | 15 (6.9%) |
| Private/NGO | 85 (38.9%) |
| Other[b] | 8 (3.7%) |
| **Marital status** | |
| single | 15 (6.9%) |
| married | 191 (87.6%) |
| divorced | 12 (5.5%) |
| **child age in month** | |
| 6–11 month | 60 (27.5%) |
| 12–17 month | 97 (44.5%) |
| 18–23 month | 61 (27.9%) |
| **Child sex** | |
| Male | 115 (52.7%) |
| Female | 103 (47.2%) |
| **Breastfeeding status (Before fasting)** | |
| Yes | 193 (88.5%) |
| No | 25 (11.5%) |

[a] student /own business

[b] student /daily laborer/own business/priest.

## Fasting characteristics of the mothers/care givers during the lent fasting period

All mothers/caregivers (n = 218, 100%) were fasting during the Lent season (also known as the Great Fast). The subjects had been practicing the fasting rituals for a mean of $17 \pm 7.2$ years. Among the respondents 89.4% of them practiced all the seven fasting periods of the Ethiopian Orthodox Church (fasting during the eves of Christmas and epiphany (Gehad tsom), the fast of the prophets (Gena tsom), the fast of Nineveh, the great fast, the fast of Apostles (Sene tsom), fast of assumption of the Virgin Merry and Wednesdays and Fridays throughout the year. During the fasting period, all of the fasting mothers/caregivers practice avoided animal source foods. In addition, 66.5% of them stayed up to 3 PM without eating/drinking any meal/drinks and 50.9% of them skipped breakfasts and stayed up to 6PM pm, and 3.6% of the fasting group fasted until 3 pm without taking any food or drink (Table 2).

## Food consumption pattern and MDD of children 6–23-months age during the non-fasting and lent fasting periods in Debrebirhan, North Shewa Ethiopia, 2019

Before the fasting period the mean Dietary Diversity Score (DDS), was 3.05 ($\pm$ 0.94) ranging between 1 and 6. During the fasting period, the mean DDS was 2.68 ($\pm$ 0.68), ranging between 1 and 6.

The proportion of children who consumed grains, roots and tubers, legumes and nuts, dairy products, flesh foods, eggs, Vitamin A rich fruits and vegetable and other fruits and vegetables in the pre-fasting period were 98.6%, 79.4%, 55.5%, 17.9%, 29.4%, 9.2% and 15.1% respectively. In the fasting period 99.1%, 82.1%, 42.6%, 0.46%, 7.3%, 18.3% and 19.7% of children have consumed grains, roots and tubers, legumes and nuts, dairy products, flesh foods, eggs, Vitamin A rich fruits and vegetable and other fruits and vegetables respectively (Fig 1).

Before the fasting period, 23.4% of children met the MDD score while only 5.5% of children meet the MDD score during the fasting period.

Those mothers who didn't fed any of the animal source foods were asked for the reason why they did not feed animal source foods in the previous day (in both pre-fasting and during fasting period) Around 57.1% of the respondents in the pre-fasting period claimed their economic status/insufficient income as a reason not to fed diet of animal origin. During the

**Table 2. Fasting characteristics of the mother/care giver at end line (during fasting period) in Debrebirhan town, North Shewa Ethiopia, 2019.**

| Characteristics | Frequency(%) |
|---|---|
| **Mothers/caregivers' fasting experience in the main fasting periods** | |
| Fast all the seven main fast periods | 195 (89.4%) |
| **Fasting experience(in years)** | |
| <10 years | 19 (8.9%) |
| 10-20years | 156 (72.9%) |
| >20 years | 39 (18.2%) |
| **Fasting characteristics** | |
| Not consuming animal source food except fish | 9 (4.1%) |
| Not consuming animal source food including fish | 206 (94.5%) |
| Not consuming animal source food including fish and fast up to 9 am | 145 (67.4%) |
| Not consuming animal source food including fish and fast up to 12 pm | 111 (51.6%) |
| Not consuming animal source food including fish and fast up to 3 pm | 8 (3.7%) |

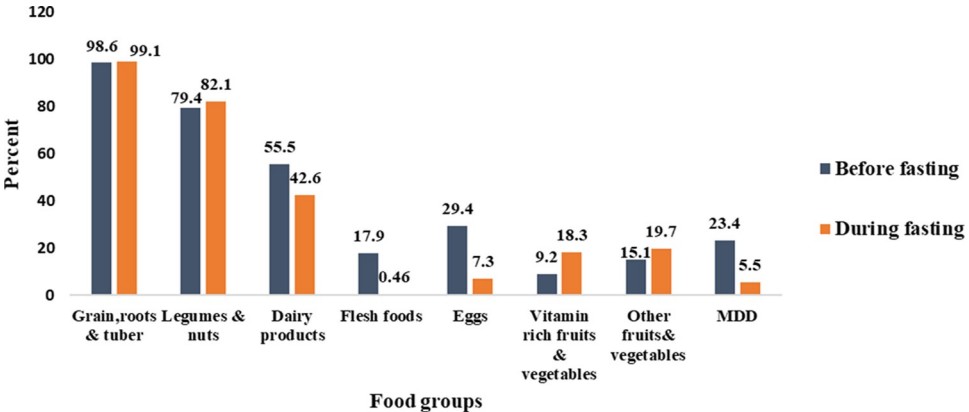

**Fig 1. Food consumption pattern and MDD of children 6 to 23 months of age during the non-fasting and lent fasting periods in Debrebirhan, North Shewa Ethiopia, 2019.**

fasting period around 79.3% of the mothers reported the fasting period as a reason not to buy and cook a non-fasting food to their children (Table 3).

## Comparison of food consumption pattern of 6–23-month-old children during the non-fasting and lent fasting periods in Debrebirhan, North Shewa Ethiopia, 2019

The result from the McNemar test shows a significant decrease in child DDS from 23.4% during the pre-fast period to 5.5% during the lent fast season (P<0.001). The proportion of children who consumed dairy product was significantly higher (55.5%) before the fasting period compared to consumption during the fasting period (42.6%) (p<0.001). Consumption of flesh food was significantly higher before the fasting period (17.9%) compared to consumption during the fasting period (0.46%) (P<0.001). Similarly, consumption of egg was significantly higher before the fasting period (29.4%) compared to consumption during the fasting period (7.3%) (P<0.001)However, there was no significant difference in the consumption of grains, roots and tubers, legumes and nuts and other fruits and vegetable food groups in the two periods (Table 4).

## Discussion

The main aim of this study was to see how fasting practices of mothers and caregivers affected the dietary diversity of infants aged 6 to 23 months by measuring their food consumption

**Table 3. Mothers/caregivers' reasons for not feeding foods of animal origin to their children (before and during the fasting period in Debrebirhan, North Shewa Zone Ethiopia, 2019.**

| Variable | Before fasting | During fasting |
|---|---|---|
| Reason not to fed diet of animal origin | Frequency (%) | Frequency (%) |
| Insufficient income | 36 (57.1%) | 4 (3.3%) |
| Non-availability | 1 (1.6%) | 11 (9.1%) |
| child is too young | 9 (14.3%) | 3 (2.5%) |
| Fasting status | 0 (0.0) | 96 (79.3%) |
| Other[d] | 17(26.9%) | 7 (5.8%) |

[d]child didn't like it/ he refuse to eat/ I don't remember

**Table 4. Comparison of food consumption pattern of 6–23-month-old children during the non-fasting and lent fasting periods in Debrebirhan, North Shewa Ethiopia, 2019.**

| Characteristics | Before Fasting Frequency (%) | During fasting Frequency (%) | McNemar P-value | 95% CI |
|---|---|---|---|---|
| Children who consumed grain, roots, and tuber | 215(98.6%) | 216(99.1%) | 1.0000 | (-0.02,0.02) |
| Children who consumed legumes and nut | 173(79.4%) | 179(82.1%) | 0.5446 | (-0.05,0.11) |
| Children who consumed dairy product | 121(55.5%) | 92(42.6%) | 0.0118** | (-0.23,-0.03) |
| Children who consumed flesh foods | 39(17.9%) | 1(0.46%) | 0.0000** | (-0.23,-0.12) |
| Children who consumed Egg | 64(29.4%) | 16(7.3%) | 0.0000** | (-0.29, -0.15) |
| Children who consumed Vitamin A rich fruits and vegetable | 20(9.2%) | 40(18.3%) | 0.0029* | (0.03,0.15) |
| Children who consumed other fruits and vegetables | 33(15.1%) | 43(19.7%) | 0.2370 | (-0.03,0.12) |
| MDD | 51(23.4%) | 12(5.5%) | 0.0000** | (-0.25,-0.11) |

P—Values were from comparison between before and during fasting using McNemar test with significant level at p < 0.05

*P-value is significant at < 0.01.

**P-value is significant at < 0.001.

patterns before and during fasting season. According to the findings, the fasting status of mothers/caregivers has a negative impact on the dietary diversity of children 6–23 months by reducing their consumption of animal source foods. We found a significant difference in dietary diversity due to changes meat, egg, milk and Vitamin A rich fruits and vegetables consumption.

In the first measurement i.e. before the fasting period, 23.4% of children met the required dietary diversity which is consistent with results from a meta-analysis pooled prevalence of dietary diversity feeding practice in Ethiopia [16] and with results from some African regions like Namibia, Benin, Ghana, Rwanda and Mozambique [17]. However, this finding is higher than the previous results conducted in different regions of Ethiopia [16, 18–21] and lower when compared with studies conducted in Kenya, Bangladesh, Nepal and Srilanka [5, 17].

This variation in results could be due to the differences in study setting. Debrebirhan is close to Addis Ababa and thus much more urbanized than the areas in the previous studies [19, 20]. Socioeconomic factors like women's literacy rate [5, 17] and the timing of different studies also vary [13, 19, 20, 22]. This finding is also different from the 2011 and 2016 Ethiopian Demographic and Health Survey (EDHS) analysis results [23, 24], and this might be explained by the fact that the EDHS was performed nationwide on a larger sample. In the second measurement during the fasting time, the children's dietary diversity significantly decreased to 5.5%. This result is in line with the result from a study done in Dejen District North West Ethiopia [13].

The diet of children in the study area was on average dominantly composed of grains (98.8%) and legumes (80.7%) which were consumed in the same amount in both the pre and intra fasting periods. This could be because this are the main staple food groups consistently consumed by Ethiopian population irrespective of fasting periods. This finding is similar to results from a study done in the Northwest and Southern region of Ethiopia [22, 25, 26].

The present study provides evidence that the children dietary diversity could be affected by the mothers/care givers' fasting status. The consumption of different food groups, specifically animal foods, significantly decreased during the fasting period. This in turn resulted in a significant decrement of child dietary diversity in the fasting period.

Before the fasting time, the consumption of flesh foods was 17.9% which is almost similar to results from studies done in Dabat, Addis Ababa and Pakistan [20, 25, 27]. But it was relatively higher when compared to the results of other studies [19, 20, 22, 26]. This could be attributed to the variation in the study time and also the possibility that the community in the

area of this study had a good animal food consumption culture [28]. During the fasting period, the proportion of children who ate flesh foods decreased to 0.5%, which is comparable with the result of a study done during a fasting period (0%) [13, 19, 26].

In the pre fasting period 29.4% of children consumed egg, which is almost identical with the report from Addis Ababa (30%) and Wolaita Zone (32.2%) [25, 29]. However, it dropped to 7.3% during the fasting period which is higher than the Dejen report (2%) [13]. This can be possibly due to the fact that most of the mothers/caregivers in our area were from urban residences and potentially had better child feeding practices than mothers/caregivers in the prior area. Before fasting, 55.5 percent of children consumed dairy products. This result was almost close to the finding from a study done in Addis Ababa [25]. During the fasting period, this number dropped to 42.6 percent, far higher than the Dejen report. This disparity may be explained by the fact that our study location has a larger cattle production and milk availability [28].

The mothers'/caregivers' fasting status is primarily responsible for the decrease in animal source food consumption during the fasting season. Despite the fact that children under the age of seven are exempt from the fasting rule, moms and caregivers did not segregate the items and feed them to their children during the fasting seasons. Fear of contamination of equipment used by other family members, to avoid the scent of the meal while preparing, to avoid breaking their fast by handling those foods (79.3%), and non-availability of those foods in the neighborhood during this fasting period are among the key reasons (9.1 percent). This finding is supported by the previous studies [13, 14, 30] which reported the mothers/caregivers' fasting practice as one of the contributing factors not to attain the recommended child dietary diversity. A study done by Seleshe et al also showed that this religious periodic restriction of animal food consumption influences meat consumption pattern in the population. The main reason for this was most of butcheries were closed due to the decrease in demand of meat during fasting seasons including Wednesday and Friday, as a result it was difficult to obtain animal foods in these periods [30].

Vitamin A rich fruits and vegetables were consumed by 9.2% of the children before the fasting time. This finding was similar to studies conducted in Amhara and Southern regions [18, 20]. As opposed to meat, milk, and egg consumption, consumption of vitamin A rich fruits and vegetables increased to 18.3% in the fasting time. This was similar to a study conducted in Dejen during the Lent fast season, where the consumption of Vitamin A rich fruits and vegetables was found to be 17.5% [13]. This might be because vegetables are more accessible and available in quantity during fasting season and mothers predominantly use those vegetables for consumption at this period.

The main strength of the study is we have conducted the survey in the pre and during fasting period, which enabled us to see the significant difference in the child dietary diversity between these two periods due to the fasting status of the mothers. The limitation of this study is we have used only a qualitative 24 hour dietary recall.

## Conclusion

We found that mothers/caregivers' fasting status or abstention from animal foods during the Orthodox fasting periods negatively affected the dietary diversity of children aged 6–23 months in the household by decreasing the consumption of animal source foods. This could be related to fear of utensil contamination as well as non or limited availability of those foods in fasting period. Therefore, the mothers/caregivers' fasting practice might be a worsening factor to the problem issue, although the dietary diversity of children aged 6–23 months in this area is already very low in general. Intervention strategies in promoting children's dietary

diversity should be designed in a way that considers Ethiopian Orthodox Tewahido Christian mothers/caregivers' fasting practice. Future nutrition studies should also consider fasting periods/respondents fasting status while measuring the children dietary diversity.

## Supporting information

**S1 Data. The dataset of the study.**
(DTA)

## Acknowledgments

We would like to thank all the study participants for their willingness and tolerance during the repeated measurement. We also acknowledge all data collectors and supervisors who contributed to the success of this research study. We are very thankful for Mrs. Heather Moorman for her cooperation and assistance with language editing.

## Author Contributions

**Conceptualization:** Addisalem Zebene Armdie.

**Formal analysis:** Addisalem Zebene Armdie, Esete Habtemariam Fenta, Solomon Shiferaw.

**Investigation:** Addisalem Zebene Armdie.

**Methodology:** Addisalem Zebene Armdie, Esete Habtemariam Fenta, Solomon Shiferaw.

**Writing – original draft:** Addisalem Zebene Armdie, Esete Habtemariam Fenta, Solomon Shiferaw.

**Writing – review & editing:** Addisalem Zebene Armdie, Esete Habtemariam Fenta, Solomon Shiferaw.

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
