## [Decision Letter · Decision Letter 0]

28 Sep 2021

PONE-D-20-37450

The effect of mothers/caregivers’ fasting status on the dietary diversity of children 6-23 months: - A longitudinal study in Debrebirhan, Ethiopia

PLOS ONE

Dear Dr. Zebene,

Thank you for submitting your manuscript to PLOS ONE. After careful consideration, we feel that it has merit but does not fully meet PLOS ONE’s publication criteria as it currently stands. Therefore, we invite you to submit a revised version of the manuscript that addresses the points raised during the review process.

The reviewers raised a number of concerns with regard to the statistical analysis, the study rationale and the interpretation of the results. Their comments can be viewed in full, below.

We look forward to receiving your revised manuscript.

Kind regards,

Natasha McDonald, PhD

Associate Editor

PLOS ONE

Journal Requirements:

A clean copy of the edited manuscript (uploaded as the new *manuscript* file)”.

3. Thank you for stating the following financial disclosure: "The funders had no role in study design, data collection and analysis, decision to publish, or preparation of the manuscript."

4. Please upload a new copy of Figure 1 as the detail is not clear. Please follow the link for more information: " ext-link-type="uri" xlink:type="simple">https://blogs.plos.org/plos/2019/06/looking-good-tips-for-creating-your-plos-figures-graphics/"
" ext-link-type="uri" xlink:type="simple">https://blogs.plos.org/plos/2019/06/looking-good-tips-for-creating-your-plos-figures-graphics/".

5. Please ensure that you refer to Figure 1 in your text as, if accepted, production will need this reference to link the reader to the figure.

Reviewers' comments:

Reviewer's Responses to Questions

**Comments to the Author**

1. Is the manuscript technically sound, and do the data support the conclusions?

Reviewer #1: Yes

Reviewer #2: Yes

Reviewer #3: No

2. Has the statistical analysis been performed appropriately and rigorously? 

Reviewer #1: Yes

Reviewer #2: Yes

Reviewer #3: Yes

3. Have the authors made all data underlying the findings in their manuscript fully available?

Reviewer #1: Yes

Reviewer #2: Yes

Reviewer #3: No

4. Is the manuscript presented in an intelligible fashion and written in standard English?

Reviewer #1: Yes

Reviewer #2: Yes

Reviewer #3: Yes

5. Review Comments to the Author

Reviewer #1: The study is an interesting one which contributes to fill the gap of studies on the effect of mothers/caregivers’ fasting status on the dietary diversity of children 6-23months. There are however some minor corrections that need to be effected by the authors.

Abstract

Line 38: remove dot in two places

Line 50: The proportion of children who met the minimum

Line 62: caregivers could be added

Introduction

Line 98: should be......….. their children are not expected to fast, it is possible

Line 105: Despite concerns over the potential negative effect

Materials and methods

Description of methods was adequate.

Line 117: Is it Woreda or Worede

Line 119: include reference for the 2018 estimates for the population

Line 120: should be………. of which 5,084 reside in......

Line 129: should be... …dietary diversity to be 27.2% and 13.6% before and during fasting period, respectively

Line 139: should be......... ..stayed up to some hours without consuming

Line 142: Include reference for the WHO IYCF indicator

Line 145: Write MDD in full first time use

Lin 147 to 148: We have interviewed should be We intervewed

Results

Lines 168 and 169:should be.....…were males and the rest were females

Line 171: One hundred and twenty-five (57.4%) mothers

Table 1: Should be Fathers occupation not Father occupation

Line 186: … all of the fasting mothers/caregivers practice avoided animal source foods

Lines 208 and 209: should be………. reason why they did not feed animal source foods in the previous day

Line 215: Table 3 Mothers/caregivers’ reasons for not feeding foods of animal origin to their children

Line 225: should be....…. shows a significant decrease in child DDS……..

Discussion

The discussion is in line with the findings.

Line 238: should be………. by measuring their food consumption pattern before and during fasting

Line 240:should be.........…. by decreasing their consumption of animal source foods

Line 241: We a significant difference in dietary diversity due to change 241 in the consumption of meat, egg, milk and Vitamin A rich fruits and vegetables

The sentence needs to be rephrased, as it is not clear what authors are trying to say.

Line 243: In the first measurement i.e. before the fasting period d, …………..

Line 271: should be…….children who ate flesh foods decreased to 0.5%, which is comparable with the result of a study done…………

Line 283: should be ..........The drop in consumption of animal……….

Line 292:……….Seleshe.S and his colleges also showed that…

Should be Seleshe and colleagues or Seleshe et al.

Line 301:should be ………where the consumption of Vitamin A rich fruits and vegetables……

Line 305: should be……..to see the significant difference in the child dietary diversity….

Line 307: The limitation of this study is we have used only a qualitative 24 hour dietary recall.

Conclusion

Line 311: Authors should note that there was no statistical analysis to show the strong relation of fear of utensil contamination as well as non or limited availability of those foods to fasting period in their findings. It is therefore suggested that this paragraph be rephrased to capture actual findings of the study.

General comments

Generally, the authors tried to describe the effect of mothers/caregivers’ fasting status on the dietary diversity of children 6-23months in the study area.

However, in few places within the manuscript, authors started sentences with figures, this should be corrected.

To improve clarity, the manuscript will benefit from language editing.

Reviewer #2: All the justifications are in place. However I think the author has to add some points to strong support the rationale for conducting this study which will add value to the the research . This will help further research on this arena.

Reviewer #3: The study still highlights a novel research topic that has broader social implications, especially for young infants whose health is sensitive to nutritional changes surrounding cultural practices. I would therefore consider the paper for publication after the authors address the comments that are major in nature. There are various issues that I have raised below for the authors to first address, which are multiple but clustered around the interpretation of the findings.

6. PLOS authors have the option to publish the peer review history of their article (what does this mean?). If published, this will include your full peer review and any attached files.

Reviewer #1: **Yes: **Ukegbu Patricia

Reviewer #2: No

Reviewer #3: **Yes: **Dr Danish Ahmad

---

## [Author Response · Author response to Decision Letter 0]

31 Oct 2021

We received funding for data collection from Addis Ababa University's School of Public Health

The funders had no role in study design, data collection and analysis, publication decision, or manuscript preparation.

---

## [Decision Letter · Decision Letter 1]

14 Dec 2021

PONE-D-20-37450R1The effect of mothers/caregivers’ fasting status on the dietary diversity of children 6-23 months: - A longitudinal study in Debrebirhan, EthiopiaPLOS ONE

Dear Dr. Zebene,

Thank you for submitting your manuscript to PLOS ONE. After careful consideration, we feel that it has merit but does not fully meet PLOS ONE’s publication criteria as it currently stands. Therefore, we invite you to submit a revised version of the manuscript that addresses the points raised during the review process. Two reviewers have re-evaluated your manuscript, and determined that their concerns were largely addressed. However, upon reviewing this manuscript myself, there remain some minor language/editing issues throughout. In addition, please also ensure that all terms which may be unfamiliar to readers are defined, such as 'woreda' and 'kebele'.  Please also take care to improve statistical reporting and report exact p-values for all values greater than or equal to 0.001, and refer to p-values as "p.001" instead of "p=.000". Our statistical reporting guidelines are available at https://journals.plos.org/plosone/s/submission-guidelines#loc-statistical-reporting.

If applicable, we recommend that you deposit your laboratory protocols in protocols.io to enhance the reproducibility of your results. Protocols.io assigns your protocol its own identifier (DOI) so that it can be cited independently in the future. For instructions see: https://journals.plos.org/plosone/s/submission-guidelines#loc-laboratory-protocols. Additionally, PLOS ONE offers an option for publishing peer-reviewed Lab Protocol articles, which describe protocols hosted on protocols.io. Read more information on sharing protocols at https://plos.org/protocols?utm_medium=editorial-emailutm_source=authorlettersutm_campaign=protocols.

We look forward to receiving your revised manuscript.

Kind regards,

Avanti Dey, PhD

Staff Editor

PLOS ONE

Journal Requirements:

Reviewers' comments:

Reviewer's Responses to Questions

**Comments to the Author**

1. If the authors have adequately addressed your comments raised in a previous round of review and you feel that this manuscript is now acceptable for publication, you may indicate that here to bypass the “Comments to the Author” section, enter your conflict of interest statement in the “Confidential to Editor” section, and submit your "Accept" recommendation.

Reviewer #1: All comments have been addressed

Reviewer #3: All comments have been addressed

2. Is the manuscript technically sound, and do the data support the conclusions?

Reviewer #1: Yes

Reviewer #3: Yes

3. Has the statistical analysis been performed appropriately and rigorously? 

Reviewer #1: Yes

Reviewer #3: Yes

4. Have the authors made all data underlying the findings in their manuscript fully available?

Reviewer #1: Yes

Reviewer #3: (No Response)

5. Is the manuscript presented in an intelligible fashion and written in standard English?

Reviewer #1: Yes

Reviewer #3: Yes

6. Review Comments to the Author

Reviewer #1: The authors have adequately addressed all comments raised in the previous review. The manuscript is technically sound and the data collected supports the findings. Appropriate statistics were carried out and manuscript was presented in comprehensible manner

Reviewer #3: (No Response)

7. PLOS authors have the option to publish the peer review history of their article (what does this mean?). If published, this will include your full peer review and any attached files.

Reviewer #1: **Yes: **Patricia Ogechi Ukegbu

Reviewer #3: **Yes: **Dr Danish Ahmad

---

## [Author Response · Author response to Decision Letter 1]

4 Jan 2022

Dear Editor,

Thank you very much for your comments on our manuscript. All of your comments were very helpful for revising and improving our paper. 

We have revised the language for some of manuscript section according to your comment. We have defined some terms which may be unfamiliar to readers and ensured that we used appropriate statistical reporting as per PLOS guideline

N:B. There is an attached file for the reviewers' comment, but we discovered that it is identical to the previous one, with all of the comments previously addressed and the feedback letter uploaded separately in the previous revision. Please let us know if we need to submit that again.

---

## [Decision Letter · Decision Letter 2]

7 Feb 2022

The effect of mothers/caregivers’ fasting status on the dietary diversity of children 6-23 months: A longitudinal study in Debrebirhan, Ethiopia

PONE-D-20-37450R2

Dear Dr. Zebene,

We’re pleased to inform you that your manuscript has been judged scientifically suitable for publication and will be formally accepted for publication once it meets all outstanding technical requirements.

Kind regards,

Solveig A. Cunningham, Ph.D.

Academic Editor

PLOS ONE

Additional Editor Comments (optional):

The title is cumbersome.  Are there both mothers and caregivers in the dataset?  If it is just mothers, you could specify that.  If it is both, than an "and" could be used for claarity.

Reviewers' comments:

Reviewer's Responses to Questions

**Comments to the Author**

1. If the authors have adequately addressed your comments raised in a previous round of review and you feel that this manuscript is now acceptable for publication, you may indicate that here to bypass the “Comments to the Author” section, enter your conflict of interest statement in the “Confidential to Editor” section, and submit your "Accept" recommendation.

Reviewer #1: All comments have been addressed

Reviewer #3: (No Response)

2. Is the manuscript technically sound, and do the data support the conclusions?

Reviewer #1: Yes

Reviewer #3: Yes

3. Has the statistical analysis been performed appropriately and rigorously? 

Reviewer #1: Yes

Reviewer #3: Yes

4. Have the authors made all data underlying the findings in their manuscript fully available?

Reviewer #1: Yes

Reviewer #3: Yes

5. Is the manuscript presented in an intelligible fashion and written in standard English?

Reviewer #1: Yes

Reviewer #3: No

6. Review Comments to the Author

Reviewer #1: The authors have adequately addressed all comments raised in the previous round of review. The manuscript is technically sound and the data supports the conclusions. The statistical analysis is appropriate for the data collected. The manuscript is therefore recommended for publication in your reputable journal

Reviewer #3: Dear authors,

While the revised paper is improved, the authors have not addressed comments adequately from previous reviews

This is now the second revision which addresses main comments from previous reviews. However, previously, reviewers provided feedback that the paper has limitations with regards to grammar, language and referencing. These issues remain uncorrected in places.

For example, lines’ 123-124 124 Children 6-23 months whose mothers/caregivers were Orthodox Tewahido Christian and who 124 were living in randomly selected households were considered as the study population’ is not.

Similarly, the authors don’t consistently use % or percentages. While % are used mostly, lines 292 and 302 have used per cent for numbers as shown below:

• 292 fasting, 55.5 percent of children consumed dairy products.

• 302 fasting period are among the key reasons (9.1 percent).

The referencing in line 142 is incorrect as ‘products. (11).’ the reference needs to come before the period.

Lastly the, reference number 2 is incorrectly cited

7. PLOS authors have the option to publish the peer review history of their article (what does this mean?). If published, this will include your full peer review and any attached files.

Reviewer #1: **Yes: **Ukegbu, Patricia Ogechi

Reviewer #3: **Yes: **Dr Danish Ahmad

---

## [Editor Report · Acceptance letter]

14 Feb 2022

PONE-D-20-37450R2 

The effect of mothers and caregivers’ fasting status on the dietary diversity of children 6-23 months: A longitudinal study in Debrebirhan, Ethiopia 

Dear Dr. Armdie:

I'm pleased to inform you that your manuscript has been deemed suitable for publication in PLOS ONE. Congratulations! Your manuscript is now with our production department. 

Kind regards, 

on behalf of

Dr. Solveig A. Cunningham 

Academic Editor

PLOS ONE